# Use of Data Mining to Predict the Influx of Patients to Primary Healthcare Centres and Construction of an Expert System

Juan J. Cubillas [1,*], María I. Ramos [2] and Francisco R. Feito [3]

1   Information and Communication Technologies Applied to Education, International University of La Rioja, 26006 Logroño, Spain
2   Department of Cartography, Geodesy and Photogrammetry Engineering, University of Jaen, 23071 Jaén, Spain
3   Department of Computer Science, University of Jaen, 23071 Jaén, Spain
*   Correspondence: juanjose.cubillas@unir.net

**Abstract:** In any productive sector, predictive tools are crucial for optimal management and decision-making. In the health sector, it is especially important to have information available in advance, as this not only means optimizing resources, but also improving patient care. This work focuses on the management of healthcare resources in primary care centres. The main objective of this work is to develop a model capable of predicting the number of patients who will demand health care in a primary care centre on a daily basis. This model is integrated into a decision support system that is accessible and easy to use by the manager through a web application. In this case, data from a primary care centre in the city of Jaén, Spain, were used. The model was estimated using spatial-temporal training data, the daily health demand data in that centre for five years, and a series of meteorological data. Different regression algorithms have been employed. The workflow requires selecting the parameters that influence the health demand prediction and discarding those that distort the model. The main contribution of this research is the daily prediction of the number of patients attending the health centre with absolute errors better than 3%, which is crucial for decision-making on the sizing of health resources in a primary care health centre.

**Keywords:** data mining; expert system; primary health care; resource optimization

## 1. Introduction

Nowadays, there are a multitude of decision support systems that organizational leaders use to make better decisions. These systems are used in numerous sectors, and many of these applications have been documented in the literature. There are also cases where significant research has been carried out using data mining techniques. Data mining is geared towards exploring data and finding solutions to particular business issues. Data mining, often also referred to as knowledge discovery in databases (KDD), is a sub-discipline of computer science aiming at the automatic interpretation of large datasets [1]. The classic definition of knowledge discovery by Fayyad et al. from 1996 describes KDD as the non-trivial process of identifying valid, novel, potentially useful, and ultimately understandable patterns in data [2]. The application of data mining in healthcare has increased in recent years because the healthcare sector is rich in information. Healthcare centres generate and collect large volumes of information on a daily basis. Despite this large volume of health and patient care information, it remains largely under-utilized [3], although it has been progressively applied in healthcare to assist clinical diagnoses and disease predictions [1]. In fact, recent studies have shown that most of the data mining work in the health sector is used in the analysis of diseases and pandemics [4]. Nevertheless, this information has been known to be rather complex and difficult to analyse.

The proper management and exploitation of health information must be used to improve patient care and optimize healthcare resources. In this sense, the use of information technology allows the automation of data mining, extracting knowledge and generating

interesting patterns of data behaviour. However, this research has remained at the theoretical level. The introduction of data mining in the production sector often results in complex applications with unfriendly user interfaces, where only the developers are able to operate them. As a result, these systems are not used. In order for these types of tools to be successful, it is essential to have a decision support system with a user-friendly interface [5–8]. However, these developments are often not implemented due to their high-cost and complex development.

Timeliness is recognized as an important characteristic of service delivery in any service setting, especially in health care. Some authors emphasize the importance of time to ensure efficiency in medicine [9]. From the standpoint of the patient, an example of the lack of efficiency is the difficulty to acquire an appointment with a physician in a timely manner, particularly at certain times of the year. This is a widespread issue in primary healthcare. Patient delays and long waiting times are one of the main problems faced by physicians and the rest of the health workforce in primary care clinics. This situation implies great difficulties for medical staff when attending to patients in a correct manner. Already in 1996, the work of Starfield [10] suggested that strong primary care resulted in higher patient satisfaction scores, lower healthcare expenses, and reduced drug prescriptions.

At the moment, dimensioning of clinic resources is usually estimated from average generalized data of previous years. Therefore, despite of the number of patients who come to the doctor's office, the available resources are always the same. For the health centre managers, this criterion of sizing resources does not lead to waiting times collapsing on most days. However, in terms of healthcare demand, the accumulation of patients in waiting rooms should be avoided as much as possible, as the risks involved are well-known. The health centre must always have the necessary resources based on patient demand. In this sense, in knowing in advance the number of patients that will go to the health centre, it would be possible to prepare the medical and human resources needed to provide a quality service.

In the first phase of the prediction, it would be necessary to determine which external factors would influence the influx of patients to health centres. In other words, what factors influence the most common pathologies treated in primary care health centres. Regarding this issue, there are many studies which perform a systematic review of influential factors on diseases. For example, Dawson et al., 2007 [11] analysed the influence of meteorological factors upon stroke incidence. Oiamo et al., 2011 [12] concluded that some levels of exposure to pollution appear to influence the utilization of health care services. According to Donaldson et al., 2012 [13] seasons influence the exacerbation characteristics in patients with chronic obstructive pulmonary disease (COPD). Further research is also working in the same direction, Ferrari et al., 2012 [14], Tseng et al., 2013 [15] or Ellis et al., 2012, [16] are other examples. The search for a relationship between external factors and the onset of certain diseases is widespread throughout the scientific community. All of these studies have as their aim the possibility to anticipate a patient's visit to the doctor and also perform a better management of the medical resources. However, seasonal, environmental and meteorological variables influence diseases differently depending on the geographical areas concerned. The first studies, such as Keatinge et al., 1997 [17], focused on this issue and the conclusions are applicable to the area over which the analysis was performed. Dawson et al. 2008 [11] research in Scotland analysed the relationship between meteorological variables and acute stroke hospital admissions. Previously, in 1996, Rothwell et al. [18] had already looked for the relationship between stroke incidence and season or temperature. They concluded that further studies were required to determine these relations for certain zones. In 2014, Cubillas et al. [19] proposed an improvement in the health system using data mining tools in order to predict the number of administrative appointments requests in health centres.

This study focuses on the last issue, with the objective of designing a model to accurately predict the number of medical attendances which primary healthcare services deal with each day. In this work the methodology of the work necessary to generate the

prediction model for a particular city is described. This methodology can be applied to other geographical areas. Firstly, to search for the main factors related to common diseases. Authors mentioned above analysed environmental and meteorological ones. This study is the first to approach modelling this relationship to a high level of precision. For this reason, data have been selected for a specific city and one primary healthcare clinic. Ellis and Jenkins 2012 [8] highlighted in their research how weekdays affect the attendance rate of medical appointments on a large scale. Considering this, the medical attendances at primary healthcare centre on weekdays, months and years, but on a small scale were also used. Additionally, the daily environmental data, such as minimum, mean and maximum temperatures, rainfall, and humidity were also considered. Levels of pollution were considered as well. This is another factor that affects health and provokes an increase in visits to the doctor, Oiamo et al., 2011 [12]. Other research in medicine has used data mining and regression models in order to analyse issues related to clinical medicine and/or access points at the different levels of health care [20,21].

Most of the previous research has looked for the relationship between some environmental and temporal factors with certain pathologies. This information helps us to make a first estimation about the number of patients who may require medical attention. In a more global context, knowing the factors that affect the assistance of patients at the health centres, regardless of pathology, provides valid information in order to make a good prediction.

The aim of this work was to predict the number of medical attendances at a primary care centre each day by using various regression algorithms. The process takes as its starting point the hypothesis that there are several variables related to the target variable, e.g., patient attendance at the health centre, the history of medical care provided each day and a series of meteorological variables. In this sense, as described above, there are several studies that corroborate this relationship. For all these reasons, data mining techniques are suitable for this type of analysis [22–26], to generate a predictive model capable of predicting in advance the number of patients attending each day. In this case, a supervised learning study was used where all the predictor variables were labelled.

This information is key to the proper management of resources and a decision support system will be developed to facilitate its use by healthcare managers. In short, this work has of main objective of (1) generating a prediction model of the number of healthcare services that will be demanded daily, and as a complement to the previous objective: (2) to nest this data mining model in an application to improve the convenience, accessibility and use of the proposed software by end users, who are the managers of the health centre. Several regression algorithms can be applied, such as linear regression, logistic regression, generalized regression model, one-class support vector machine (SVM), etc. In this study, a priori the nature of the variables is unknown and little training data is available (daily health care data over the last five years were available); therefore, linear and non-linear linear regression algorithms were used.

The first algorithm selected In this study was the generalized linear model (GLM), which works mathematically as the weighted sum of the features with the mean value of the distribution assumed using the link function g, which can be chosen flexibly depending on the type of result.

$$g(EY(y|x) = \beta_0 + \beta_1 x_1 + \cdots \beta_p x_p \tag{1}$$

Another linear algorithm selected was SVM, which has the advantage of being able to be used with different kernels. These allow the data to be distributed on a hyperplane according to a function, which facilitates the adaptation of the algorithm to the nature of the data, thus allowing an infinite transformation. When using the linear kernel, the following transformation is performed:

$$K(x,x') = x \cdot x' \tag{2}$$

This algorithm fits very well if the nature of the data is linear and there are many predictor variables. It should be noted that in this algorithm there is no upper limit on the number of predictor attributes, it is just those imposed by the hardware. In this study,

there was a limited set of training data, since authors had a limited number of years with daily data of medical attendances in the health centre and few meteorological variables were needed.

The nonlinear algorithm applied was the SVM algorithm with a Gaussian kernel. This kernel applies the following transformations to the data:

$$K(x, x') = \exp(-\gamma \|x - x'\|2) \tag{3}$$

The value of $\gamma$ controls the behaviour of the kernel. If it is very small, the final model is equivalent to that obtained with a linear kernel, but if the value increases the data becomes more distant, forming a Gaussian bell in the hyperplane, adapting very well when the data nature has no linear distribution.

In summary, these are the properties of the three algorithms applied in this study based on the hypothesis of a priori ignorance of the relationship between the variables considered and the target attribute, and also considering that the training data available was limited and the predictor variables were few in number. Thus, the complexity of these algorithms means that the relationship between the attributes used cannot be described by a specific equation.

**New Contribution**

The number of patients coming to the health centre varies greatly from day to day depending on the month of the year. The main factors influencing this are weather and environmental factors. Nowadays, meteorological systems provide us with weather information in advance. In this sense, our contribution is to take advantage of this available data and generate an accurate predictive system that informs health centre managers about the number of patients that will attend the health centre. To make this algorithm a useful and user-friendly tool for management, an expert system was developed that integrated the predictive algorithms that provided the best results.

## 2. Methods

### 2.1. Dataset, Integration and Management

The methodology followed in this work was sequenced in several phases, from the data understanding, data preparation, and generation of the models to the validation and implementation of the models. The data used in this research study corresponds to the city of Jaen, Andalusia, in southern Spain. The scope of the study through the choice of primary care centres in the District of Jaén is due a collaboration agreement that was signed between the University of Jaén and the Jaén Health District. This collaboration included the cession of data of a primary healthcare centre. It covered the number of patients treated each day at primary care centres during the years 2014–2018. In addition to this information, factors that may influence the increase or decrease in the number of people requiring primary care was also considered in the study. Some environmental values recorded in the area during this time for each day were included. All the data have been classified into three groups:

Season data: date and number of visits to the primary care centres. These data are provided by the health centres in Jaen.

Meteorological data: The meteorological data used comes from two weather stations placed in the city of Jaen, Red de Información Ambiental de Andalucía (REDIAM) [27]. In order to integrate this information into the database daily mean values were taken for maximum, mean and minimum temperatures, measured in degrees Celsius, humidity and rainfall.

Pollution levels: This information was provided by its source with values termed either good or bad, depending on the partial index for each pollutant: sulphide dioxide $SO_2$, particles, $NO_2$ nitrogen dioxide, carbon monoxide $CO$ and ozone $O_3$. These data are available online at REDIAM [27].

All these data were used to create a model for predicting the number of daily requests in primary healthcare. This predictive model was generated from a retrospective study of data from 2014–2017. Data for 2018 was used to validate the resulting model. Methods for

determining the validity of regression models include comparing model predictions and coefficients with theory, collecting new data to test model predictions, comparing results with theoretical model estimates, and data splitting or cross-validation, in which part of the data is used to estimate model coefficients and the rest of the data is used to measure the accuracy of the model prediction [28]. In this case, the latter was chosen. In a first analysis, the year 2018 was used for a first validation; however, in order to definitively select the algorithm that best fits the data, a cross validation was carried out with the data from each year.

Before starting the procedures for obtaining the predictive model it was necessary to perform a previous analysis to determine data patterns. This allowed us to define the best way to group patients in order to conduct proper integration and management of all of them. In the case of data about the number of visits to health centres during 2014–2017, there was a certain similarity in the distribution of the data based on the time of year, Figure 1. The graph shows a general tendency of the distribution function, very similar from one year to another. Overall, in holiday periods in summer and winter, August and December and January, a clear reduction was detected in the amount of visits to the primary care physician. The number of visits to the doctor was variable from some months to others. Figure 1 shows a very significant difference between the months of August and October. The demand for the first month to the second dropped by 50%. This may be due to the fact that August corresponds with the holiday period.

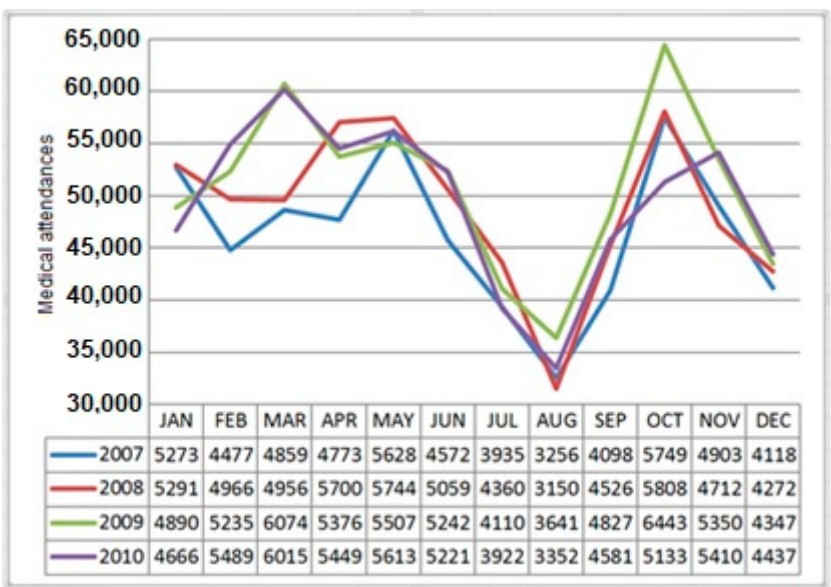

**Figure 1.** Distribution of medical attendances grouped by month from 2014 to 2017.

However, other variations that are not so obvious are displayed, for example in 2017, there was a difference of 9.2% between October and November. In other years the number of patients during these two months was very similar and yet in 2017 was very different. These differences reinforce the need to create data mining models that are able to predict these differences in behaviour.

At a higher level of detail, the influx of patients each day of the week was analysed. In this sense, the number of visits on weekdays (working days), holidays and weekends were considered. Instead of grouping the number of visits per month they were grouped by days of the week, and there was a very similar distribution from 2014–2017, Figure 2. As the week progressed the number of appointments decreased up to the weekend in which they were dramatically reduced. As noted in previous paragraphs this reduction is due to the fact that at the weekend only emergencies are attended to. For this reason, it was best for our study to separate the data in two independent cohorts:

- Holidays cohort: The dataset contained information only during holidays. This cohort contained data from Saturdays, Sundays and holidays. Only on these days, are emergencies attended to at health centres.
- Weekdays cohort: The dataset contained information only on weekdays.

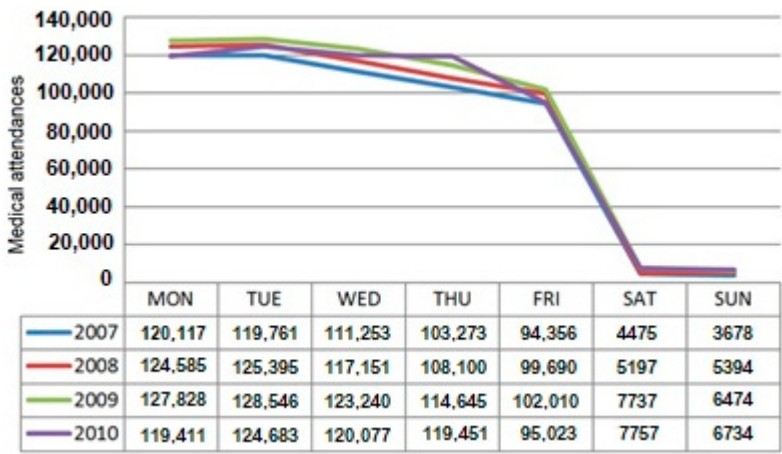

**Figure 2.** Distribution of medical attendances grouped by days of the week from 2014 to 2017.

The primary data were the date and the number of medical visits, as well as the meteorological data. When analysing Figure 2, the behaviour of the visits was very different for each day of the week (Monday, Tuesday, Wednesday, etc.), which is why the date is denoted as follows:

Thus, in view of the similarities found in the distribution of data the following groupings were considered most appropriate:

- Day of the week (Monday, Tuesday, . . . Sunday);
- Month (1 is January, 2 is February, . . . 12 is December);
- Year (2014, 2015, 2016, 2017).

Table 1 shows an example of how the data were stored before being used by the predictive models. In this case, four fields of the database were shown once all the information had been grouped and integrated.

**Table 1.** An example of training data.

| Month | Year | Day of the Week | Type of Day | Mean Temp | Max. Temp | Min. Temp | Rainfall | Rel. Hum | Air Quality | Mean Number of Patient Visits |
|---|---|---|---|---|---|---|---|---|---|---|
| 2 | 2017 | MON | W | 10.4 | 13.0 | 7.8 | 0.2 | 66.39 | 1 | 2651.75 |
| 3 | 2017 | TUE | W | 10.4 | 7.8 | 4.3 | 1.4 | 58.15 | 0 | 2563.8 |
| 7 | 2014 | SAT | H | 27.7 | 33.5 | 20.3 | 0 | 27.1 | 0 | 2228 |
| 1 | 2017 | TUE | W | 9.38 | 13.0 | 4.2 | 0.4 | 79.42 | 0 | 2112.75 |

*2.2. Data Mining Algorithms and Decision Support System Development*

As mentioned above, the 2018 data was used to validate the model. As the current objective was to develop a decision support system, the data from 2018 was used to generate the model, that is the baseline of the validation, which will give us a more solid model as it will be generated with the knowledge of the years 2014–2017.

For the development it is used the relational database management system (RDBMS) Oracle 19 Express Edition [8], which has a free licence. This tool provided everything needed for the rapid development of the decision support system: the database to store the information, Oracle data mining which is a module integrated in the database with which the data mining algorithms are designed and Oracle applications express (APEX) which is a WEB application development module that allows the development of WEB applications for both PC and mobile devices. Figure 3 shows the design of the decision support system divided into layers.

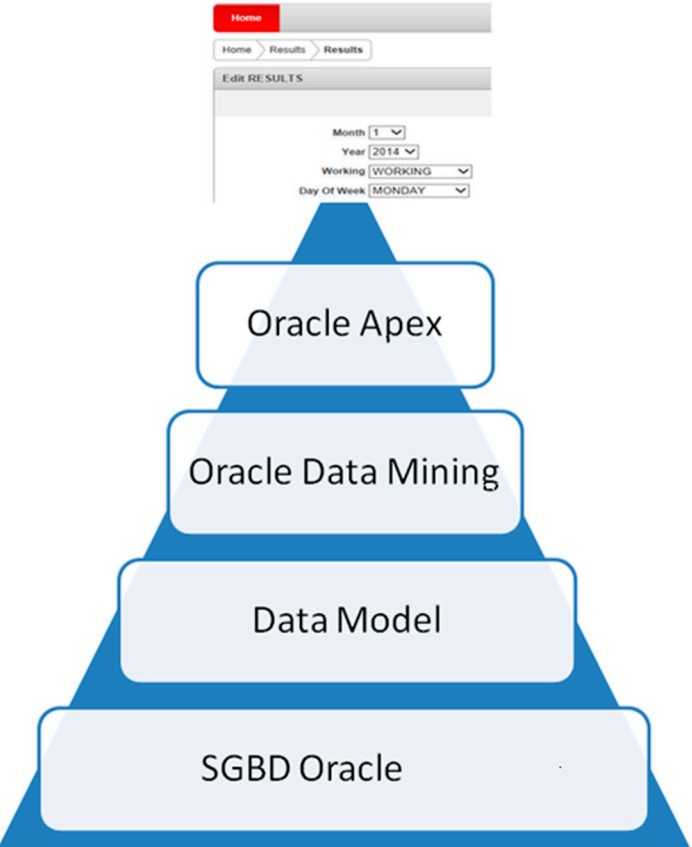

**Figure 3.** Decision support system architecture.

Figure 3 shows the structure of the modules that Oracle integrates and that were used to develop the decision-making system, at the base is the database that allowed the grouping and analysing of the root information. Then, once this process was done, the data mining study and the analysis of the results were performed with the Oracle data mining module. Finally, the Oracle application express module was used to develop the interface and invocation of the system. The development was agile since we were constantly working with the modules within the database management system.

## 3. Development of the System

It was necessary to follow a series of steps to achieve the development of our decision support system. These steps are as follows: design of the data model, upload data to generate the predictive models, generate the predictive models with Oracle data mining, develop the decision support system interface with Oracle application express, and then design and develop the system learning using feedback data.

### 3.1. Design of the Data Model

The data model needed for the system was very basic, needing just two tables. The first table contained historical data of doctor attendance, meteorological data and environmental quality, this table was used to generate our predictive model. The second table stored the results and input parameters of the prediction. Table 2 shows the attributes of the tables needed for the functioning of the decision support system.

**Table 2.** Design decision support system tables.

| HIST_TABLE_GENERATION_MODELS | RESULTS |
| --- | --- |
| Day_of_week (varchar2(20)) | Date (date) |
| Month (number) | Id_record (number) |
| Year (number) | Day_of_week (varchar2(20)) |
| Working (varchar2(1)) | Month (number) |
| number visits (number) | Year (number) |
| Minimum_temperature (number) | Working (varchar(1)) |
| Mean_temperature (number) | Minimum_temperature (number) |
| Maximum_temperature (number) | Mean_temperature (number) |
| Relative_humidity (number) | Maximum_temperature (number) |
| air_quality (number) | Relative_humidity (number) |
| | air_quality (number) |
| | Prediction (number) |

*3.2. Upload Data to Generate the Predictive Models*

The data used to generate the model were the same for the study. There are multiple tools on the market that can be used to upload data into a database. The SQL-Loader was used, which is a tool that allows to read text files and transfer the data into the tables. Once the information had been uploaded into the auxiliary tables the next step was to group the information using the framework criteria of the study:

- Day of the week (Monday, Tuesday, . . . Sunday)
- Working day (Weekday, Holiday)
- Month (1 is January, 2 is February, . . . 12 is December)
- Year (2014, 2015, 2016, 2017)

The grouped data provided the basis to generate the decision support system. The information was grouped with a simple SQL query. The query integrated all data and grouped them by the established criteria (month, year, working day). All the data were obtained from that date. See the Supplementary Materials which includes the source SQL that makes the inclusion of grouped information.

Our decision support system was based on a theoretical study carried out by the University of Jaen and the Healthcare District. The data mining study was conducted using the following input data:

Season data: date and number of visits to the primary care centres.

Meteorological data: The meteorological data used came from two weather stations in the city of Jaen [27]. In order to integrate this information into the database the daily mean values were taken for maximum, medium and minimum temperatures, humidity and rainfall.

Pollution levels: This information was provided by its source denoted as good or bad, depending on the partial index for each pollutant: sulphide dioxide $SO_2$, particles, $NO_2$ nitrogen dioxide, carbon monoxide $CO$ and ozone $O_3$. These data were also obtained from the network REDIAM [27].

The study was conducted using data from 2014 to 2018. The model was generated with data from 2014–2017. Finally, the data from 2018 was used to validate the effectiveness of the models.

Using the minimum description length algorithm (MDL) [29], the weight of each attribute over the target attribute (number of patients) was measured. This algorithm detected which attributes were not decisive for the study, and they were subsequently eliminated. The implementation of this algorithm returned a range of values between $-1$ and 1.

A higher positive value means that the attribute is more closely related to the attribute to predicted; however, a value of 0 means that there is no relationship, finally a negative value means that the attribute is not related to the target attribute and therefore may cause

noise in the study. In this study only those attributes with weights greater than 0 were considered, discarding all those with 0 or negative values.

In the first part of the study the category discrimination method (CDM) algorithm was used. Table 3 shows the result and analyses the level of influence of the attributes on the target attribute.

**Table 3.** Weight assigned to each attribute.

| Attribute Name | Cohort Weekday | Cohort Day Weekend |
|---|---|---|
| Month | 0.34 | 0.12 |
| Minimum temperature | 0.28 | 0.05 |
| Maximum temperature | 0.21 | 0.03 |
| Mean temperature | 0.25 | 0.04 |
| Day of the week | 0.06 | 0.37 |
| Relative humidity | 0.11 | 0.09 |
| air quality | 0.05 | 0.02 |
| Rain fall | −0.06 | −0.12 |

In this case, the target attribute is the number of patients requiring medical care in the health centre.

According to the MDL algorithm attributes influenced differently on each cohort. Thus, for the cohort of the weekdays the attributes which were most influential were, maximum, minimum and average temperatures, relative humidity, day of the week and air quality. However, for the cohort of the holidays the main weight was the day of the week and then, to a lesser extent, the month, maximum, minimum and average temperatures, and relative humidity. In both cases the rain had no influence on any of the target attributes.

The most important thing about the system was the correct choice of the data mining algorithms for the models. The GLM [30] regression algorithm was used for the cohort of working days data. This algorithm had an absolute error of 2.11% when compared to the prediction results with the actual data of 2018. The GLM algorithm is suitable for predictions where the target distribution is likely to follow a non-normal distribution, such as a multinomial or Poisson distribution. Similarly, GLM is useful when the relationship of the variables is not linear. The distribution of the variables and the prediction can be seen in Figure 4. Here, it is shown in a single graph the relationship between the predictor variables and the target attribute, i.e., the number of medical attendances at the health centre. Two axes are used for its representation: The first vertical axis indicates the temperature in degrees Celsius and the percentage of relative humidity. Additionally, the second indicates the number of medical attendances. The figure shows a direct relationship between relative humidity and the number of patients who visited the health centre, and an inverse relationship with temperature, i.e., the higher the temperature, the lower the number of visits. Furthermore, the figure shows that air quality had no relationship with the target attribute.

In the case of non-working days, the best performing algorithm was SVM with linear kernel [31], which had an absolute error in prediction of 5.4%. This algorithm performs well on data sets that have many attributes, even if there are very few cases on which to train the model. There is no upper limit on the number of attributes; the only constraints are those imposed by hardware. Traditional neuronal networks do not perform well under these circumstances. In our case, the non-working day cohort was much lower than that of the working days because normally one week has two non-working days and five working days, and this problem was compounded when holidays fall on weekdays. For this type of data, the training data was minimal. Figure 5 shows the relation between the number of visits with the attributes involved in the study.

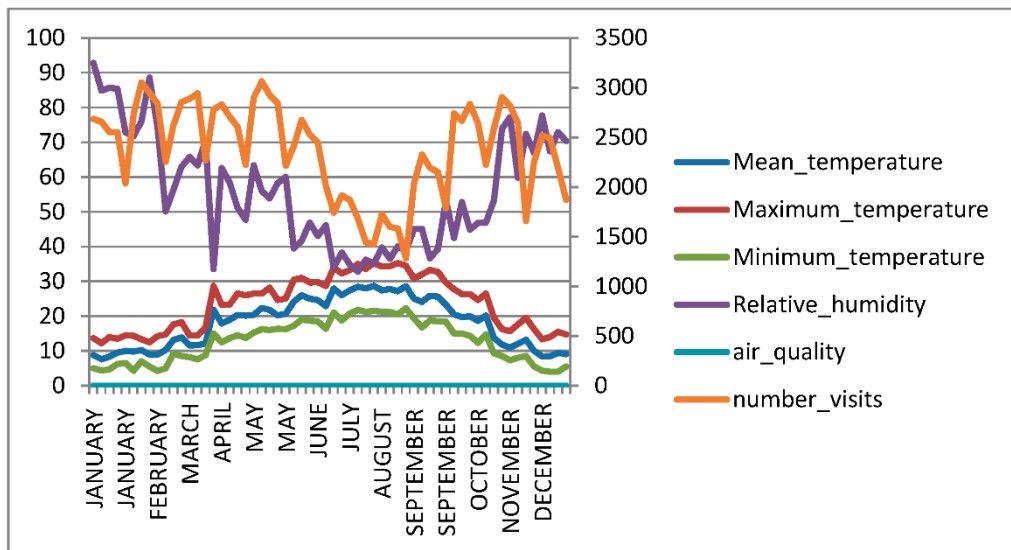

**Figure 4.** Relationship between the attribute target and predictors for working days.

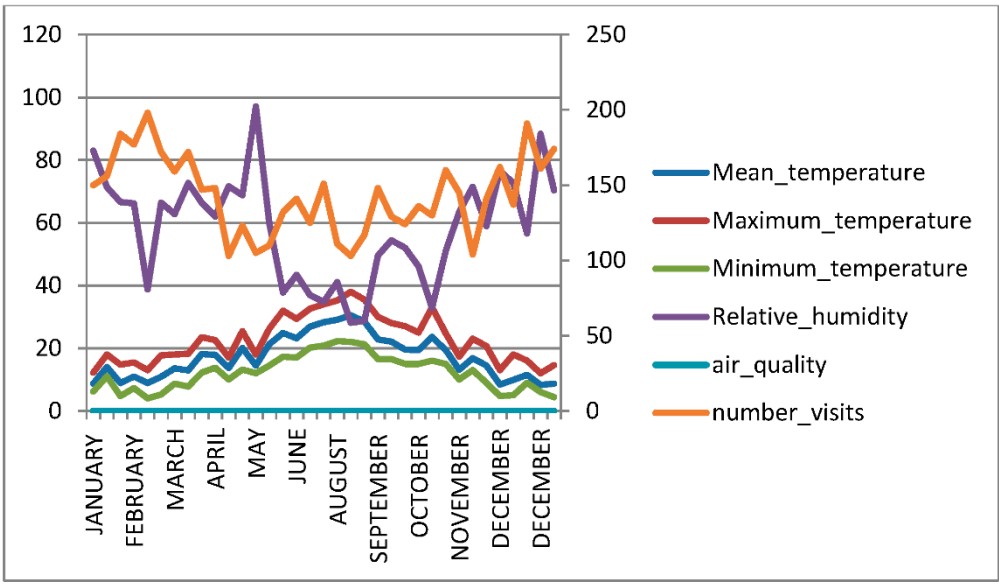

**Figure 5.** Relationship between the attribute target and predictors for holidays.

In general, the relationship between weather variables and the number of medical attendances had a different trend on public holidays than on working days. In contrast to what can be seen in Figures 4 and 5 confirms that patients who came to the health centre on public holidays were usually for emergencies, which means that weather conditions did not influence the patien's visit to the health centre.

### 3.3. Generating the Models with Oracle Data Mining

Oracle data miner is an extension that runs within the database management system, accessed through the Oracle SQL Developer tool that provides the data analyst module, running and applying data mining models. These generated models are easily accessible using a simple SQL statement within the database.

Once data are stored, models are generated by Oracle data mining. To do this the SQL Developer tool is used. All the models were generated in the same way with SQL-Developer. The table of origin (HIST_TABLE_GENERATION_MODELS) is selected. Then, a filter to distinguish between working and non-working days is applied, and then the type of model to generate is selected, in this case a regression. Finally, the last step

is to select the algorithm (GLM for working days and SVM for non-working days), and the system prompts to select the target attribute and the attributes that form part of the predictive model. The example shows the generation of the model for total visits in a non-working day. The right side of Figure 6 shows the model generation tool for data mining, shown as a visual tool.

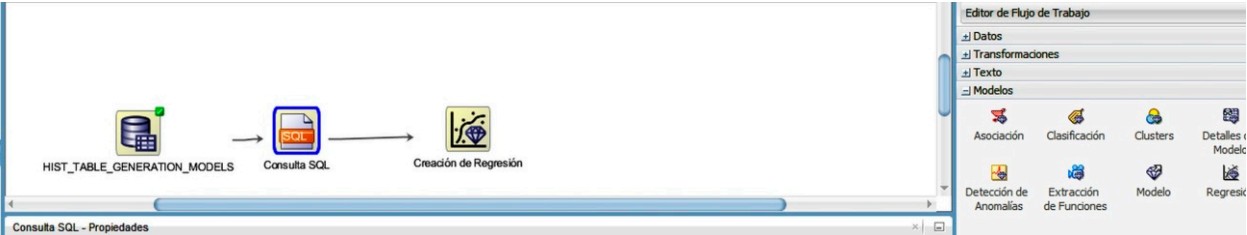

**Figure 6.** Tool for the design and development of the predictive models.

On the right side of Figure 6 the data mining models can be chosen, tables of origin, etc. On the left there is the area where the elements can be dragged that form part of the predictive model (in this case the source of Table 3, an element of the type of query SQL and the regression model). Finally, to generate the model the user only needs to click on run.

Once the model has been generated with Oracle data mining all that is needed is a simple SQL. See the Supplementary Materials to consult the syntax used.

### 3.3.1. Development of the Application with APEX

Oracle application express (Oracle APEX) is Oracl's primary tool for developing Web applications with SQL and PL/SQL. Using only a web browser, one can develop and deploy professional Web-based applications for desktops and mobile devices. To develop an application with APEX, one just has to open a web browser to connect to it. Once inside the development area, one can access various wizards that generate automatic web pages of types, REPORT and FORMS, based on the tables of origin. By selecting the table, the FORM type pages are created with all the table fields and buttons to insert, modify and delete the record. In Figure 7 several screens of the wizards that generate web pages are shown.

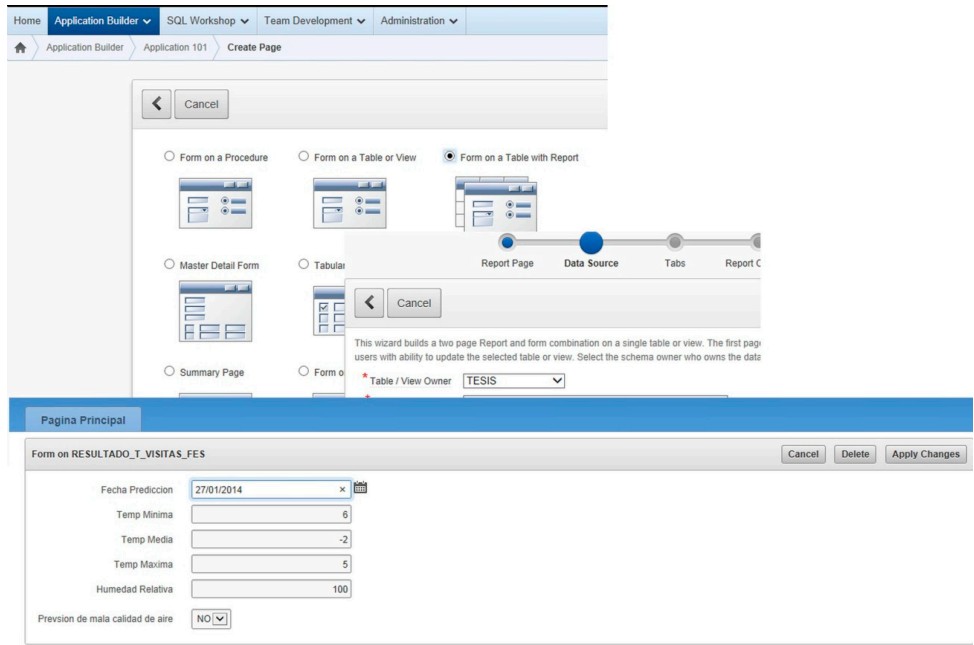

**Figure 7.** Assistant to develop web pages.

Once the data manipulation interface is generated, the invocation of the data mining model is developed. To do this, a trigger in the RESULT table is programmed so that each time an insertion or modification is made to a record, it invokes the model, and the result is stored in the "Prediction" field. The trigger code can be seen in the Supplementary Materials.

Finally, another screen to display the result of the decision support system was designed. To do this a wizard was used again relying on the same table of results. To improve the interface and assist the manager, a graph is generated in which historical data are shown, which helps to build the model (number of patient visits in previous years on a similar day). To do this a graph type component is needed and a scheduled SQL to group the data. The SQL that would generate the graph is shown in the Supplementary Materials.

3.3.2. System Learning through Data Feedback

It was essential for our decision support system to learn [19] so that it adapts to changes in the use of healthcare services. To do this, they are published in a field in the application in which the user can manually insert the actual data of patients which attended the healthcare centres, which ensures inductive supervised learning. Invoking a SQL statement, the models are generated with actual data inserted, Figure 8. The syntax to regenerate the model again can be seen in the Supplementary Materials.

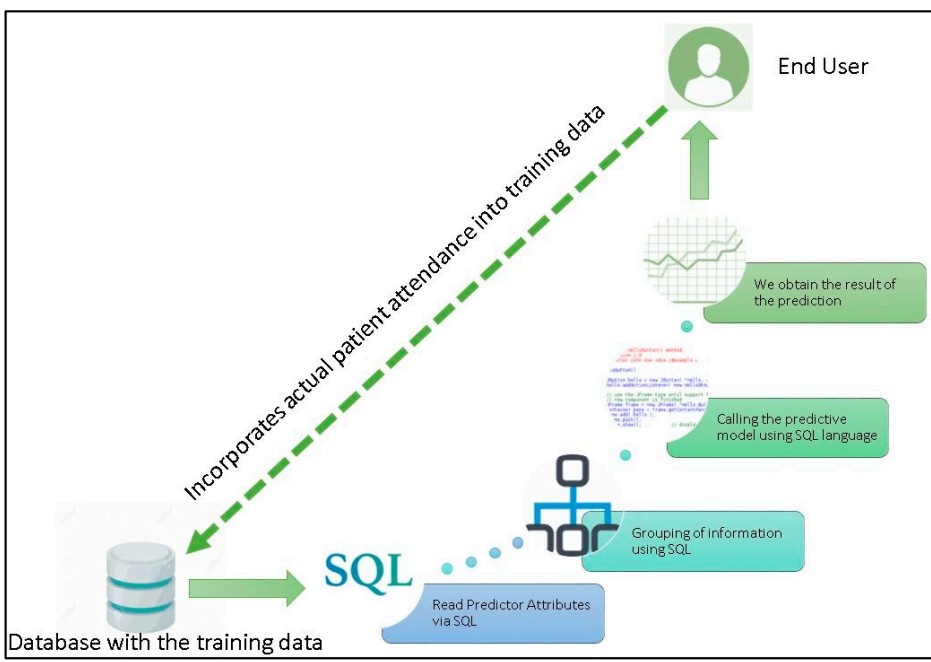

**Figure 8.** Expert system feedback from training data.

**4. Results**

The results of the first phase of the study correspond to the implementation of the MDL algorithm to identify which attributes which most influence the target attributes; in this case they were calculated for each of the two cohorts, Table 4.

**Table 4.** Results of the main metrics after generating models for the cohort of weekdays.

|  | Predictive Confidence | Mean Absolute Error | Mean Actual Value | Mean Predictive Value | Root Mean Square Error |
|---|---|---|---|---|---|
| SVM with Gaussian Kernel | 68.75% | 102.72 | 2312.25 | 2303.36 | 140.67 |
| SVM with Linear Kernel | 83.23% | 52.25 | 2312.25 | 2307.38 | 68.2 |
| GLM | 85.17% | 49.09 | 2312.25 | 2308.98 | 66.77 |

According to the MDL algorithm attributes influenced differently on each cohort. Thus, for the cohort of weekdays the heavier weighted attributes were: maximum, minimum and average temperatures, relative humidity, day of the week and air quality. However, for

the cohort of holidays the main weight was the day of the week and, although to a lesser extent, the month, maximum, minimum and average temperatures, and relative humidity. In both cases the rain had no influence on any of the target attributes.

### 4.1. Weekday Model Generation

The regression models were generated from the algorithms already described in the previous section. For the cohort of weekday, the rain attribute has been dismissed as non-influential on the target. The results obtained are shown in Table 4 where the precision of models generated from each algorithm is shown. These results concluded that the algorithm theoretically best modelled to our issue was the GLM, with a predictive confidence of 85.17% and an absolute error of 49.09.

Then, the three models were validated by comparing their predictions with the actual data of 2018. In order to perform this validation, the absolute error of the model was calculated, considering this as the difference between the predicted data and the actual data. The GLM algorithm results had an error of the 2.11% versus the 2.31% of the SVM with a linear kernel and 4.22% for the SVM with a Gaussian kernel. Figure 9 shows the distribution of data making the prediction model with the actual data. The similarity between both distributions is very significant. The good predictive ability of the GML algorithm to predict the number of patients attending the health centre on weekends and holidays was confirmed. The graph shows a generally good fit despite the large variation in the number of visits between the winter and summer months. However, a more detailed analysis of the graph showed a greater disparity in the month of October and another between the months of March and April. This may be due to the fact that the flu vaccination campaign usually starts in October and this factor was not taken into account directly as another factor in the training data. As for the months of March and April, the period for prescribing medication to treat the symptoms of spring allergies usually begins here, a fact that was not been explicitly considered as a factor, although it was indirectly included in this increase in the number of visits to the health centre.

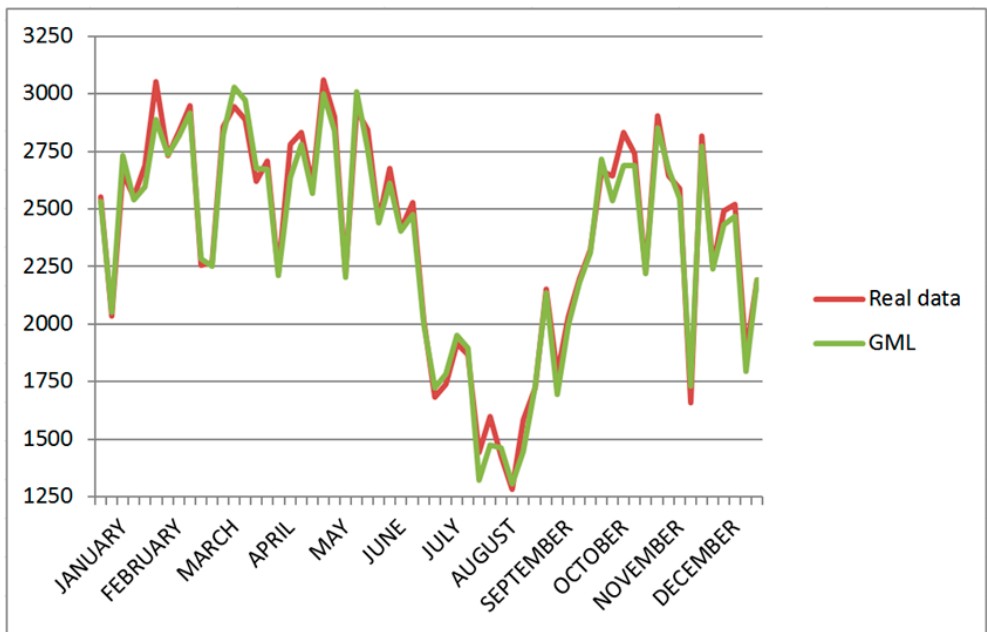

**Figure 9.** Comparison of the predictions of the weekdays with respect to the actual data of 2018.

The behaviour of the model against the action of the different factors that influence it was determined by the coefficient statistics for linear regression. This allowed us to understand what is called the logic of the model. In multivariate linear regression, the regression parameters are often referred to as coefficients. When a multivariate linear regression model is built, the algorithm computes a coefficient for each of the predictors

used by the model. The coefficient is a measure of the impact of the predictor x on the target y. Numerous statistics are available for analysing the regression coefficients to evaluate how well the regression line fits the data.

An analysis of the data for days of the week can be seen in Table 5, where the amount of patients attended on Wednesdays, Thursdays, and Fridays, and the amount of returning patients to health centres on Monday added up to a large number of patients. This is because during the weekend the health centre only provides emergency services and if a patient is ill over the weekend and the illness is not serious, they are advised to go to the health centre on Monday.

**Table 5.** Coefficient statistics for linear regression of the predictive model.

| Principio del Formulario Attribute Final del Formulario | Value | Standardized Estimate of the Coefficient | Linear Coefficient Estimate | Limit Lower Coefficient | Limit Higher Coefficient |
|---|---|---|---|---|---|
| Intercept | | 0 | 1689.13 | 1205.85 | 2172.41 |
| Day of Week | Friday | −0.432 | −469.539 | −554.58 | −384.503 |
| day of week | Thursday | −0.173 | −192.036 | −277.634 | −106.439 |
| day of week | Monday | 0.108 | 118.722 | 35.165 | 202.279 |
| day of week | Wednesday | −0.093 | −105.912 | −193.92 | −17.903 |
| relative wet | | 0.115 | 2.937 | 0.078 | 5.795 |
| month | 8 | −0.61 | −891.463 | −1025.33 | −757.594 |
| month | 7 | −0.368 | −672.808 | −828.17 | −517.446 |
| month | 2 | 0.356 | 548.605 | 294.477 | 802.733 |
| month | 10 | 0.314 | 513.201 | 345.273 | 681.13 |
| month | 11 | 0.262 | 479.058 | 227.502 | 730.615 |
| month | 4 | 0.242 | 353.385 | 176.6 | 530.169 |
| month | 5 | 0.213 | 311.587 | 165.538 | 457.636 |
| month | 1 | 0.213 | 406.81 | 118.21 | 695.41 |
| month | 3 | 0.185 | 371.092 | 131.712 | 610.473 |
| month | 9 | −0.171 | −255.791 | −382.032 | −129.55 |
| month | 12 | 0.096 | 144.521 | −137.397 | 426.439 |
| air quality | 1 | 0.068 | 102.302 | −12.32 | 216.924 |
| Maximum temperature | | −0.168 | −9.836 | −37.893 | 18.222 |
| Average temperature | | 0.752 | 48.836 | 0.738 | 96.933 |
| Minimum temperature | | −0.175 | −13.38 | −46.04 | 19.27 |

In this sense, an analysis of the attribute of the month detected that during the months of July, August and September, the number of patients decreased. Logically this is because the holiday period and high temperatures coincide. This leads to an apparent absence of pathologies. On the other hand, February and October are the months with the highest number of patients admitted to health centres.

Considering the environmental data, the important weight that the maximum, minimum and average temperature have in the model was confirmed, with a standard coefficients estimated of −0.175, 0.752 and −0.168, respectively. In terms of relative humidity is was proved to be the least significant meteorological attribute.

Finally, the quality of air was the element of the model with least influence, with a coefficient statistic estimated of 0.068. However, during the days with poor air quality it became an important attribute, with a value of estimated coefficient of 102.32.

### 4.2. Weekend and Holidays Model Generation

In the case of the cohort of holidays and weekends the regression models were generated using each of the algorithms used for the cohort of days of the week. However, the results were different, for these data the algorithm that statistically best fitted the prediction

was SVM with a linear kernel with an absolute error of 12.77, Table 6. This table shows the results of the statistical and theoretical parameters of the application of the algorithms for the holiday data cohort. Theoretically the smallest mean square error occurred with the SVM algorithm with a linear kernel. This result was logical since the volume of training data used with this cohort was smaller than in the case of the weekday cohort.

**Table 6.** Results of the main metrics after generating models for the cohort of holidays.

| | Predictive Confidence | Mean Absolute Error | Mean Actual Value | Mean Predictive Value | Root Mean Square Error |
|---|---|---|---|---|---|
| SVM with Gaussian Kernel | 2.91% | 26.27 | 132.34 | 132.28 | 34.78 |
| SVM with Linear Kernel | 39.04% | 12.77 | 132.34 | 130.66 | 21.81 |
| GLM | 35.53% | 15.72 | 132.34 | 131.3 | 23.06 |

Then, the theoretical accuracy of the model was analysed by determining the extent to which the predicted values corresponded to reality. Similarly, in the case of the cohort of weekday values predictions were compared with the actual values of patients on holiday in 2018. In this case, it was confirmed that SVM with a linear kernel was the best algorithm with an absolute error of 7.6% compared with 9.43% of the error of the GLM or 13.60% for the SVM with a Gaussian kernel.

A visual analysis of these results is shown in Figure 10 which shows how both functions are adjusted. It was observed that in some months both functions matched almost exactly while other months they did not, but the trend of both was adjusted with high precision.

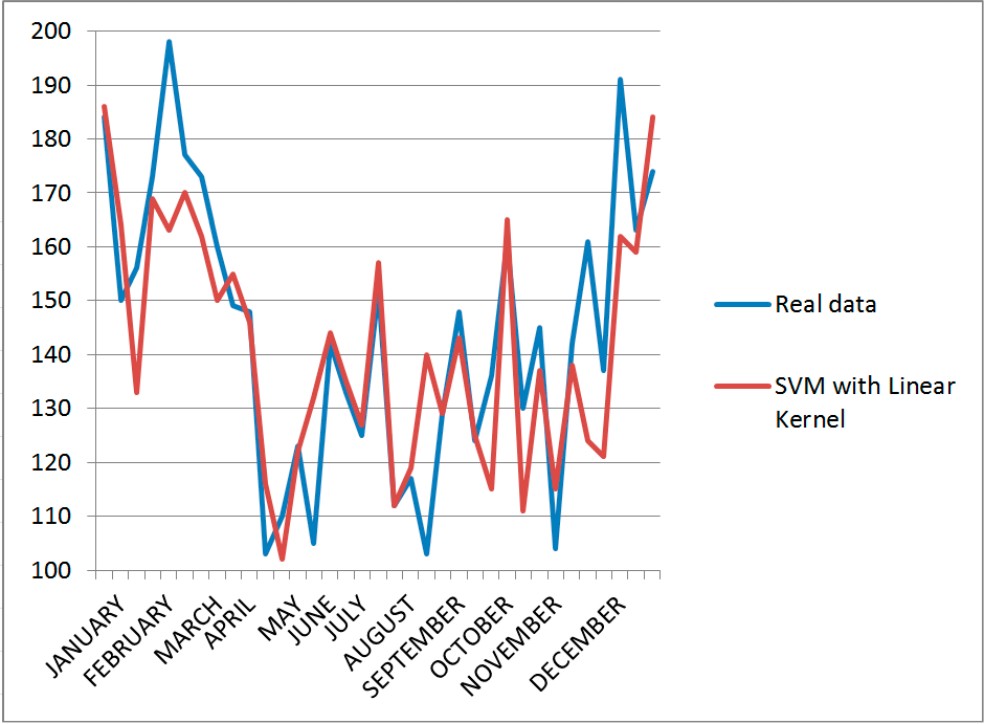

**Figure 10.** Comparison of the predictions of the holidays with respect to the actual data of 2018.

The coefficient statistics for linear regression in Table 7 was checked, facing the same external factors, and this model had a different behaviour with respect to the previous cohort of weekdays.

**Table 7.** Coefficient statistics for linear regression of the predictive model.

| AttributeFinal del Formulario | Value | Coefficient |
|---|---|---|
| Intercept | | 0.49863636 |
| day of week | MONDAY | 0.18892462 |
| day of week | TUESDAY | −0.15558891 |
| day of week | SATURDAY | 0.10970659 |
| day of week | WEDNESDAY | −0.09673843 |
| day of week | FRIDAY | −0.07478812 |
| day of week | THURSDAY | 0.02517979 |
| day of week | SUNDAY | 0.00330445 |
| relative wet | | −0.01722341 |
| month | 8 | −0.28251676 |
| month | 7 | −0.24222796 |
| month | 2 | 0.18287157 |
| month | 1 | 0.14417783 |
| month | 3 | 0.12638826 |
| month | 9 | −0.10298506 |
| month | 12 | 0.07631651 |
| month | 11 | 0.04979944 |
| month | 10 | 0.03127474 |
| month | 5 | 0.01690143 |
| month | 4 | 0 |
| month | 6 | 0 |
| air quality | 1 | 0.03298887 |
| air quality | 0 | −0.03298887 |
| maximum temperature | | 0.06454707 |
| average temperature | | 0.01384894 |
| minimum temperature | | 0.04531514 |

In an analysis of data predicted for the cohort of the weekdays it was determined that Monday was the day more patients requested the emergency services, with a coefficient of 0.188. This fact seems logical since on Saturday and Sunday primary care services are not available and therefore many patients on Monday are forced to go to the emergency department for pathologies that worsened during the weekend. On the days of the weekend, it was detected that on Saturday there was a greater coefficient value than on Sunday. This may be because on Sunday many sick users prefer an appointment scheduled for Monday in primary care. It was concluded that the most efficient algorithm in the case of working days was the GLM, whereas in the case of public holidays it was the SVM algorithm with a linear kernel. It was shown that there was a direct relationship between environmental factors and the influx of patients to the health services in Jaén and this relationship was statistically quantified with regression coefficients. The analysis of these coefficients allowed a better understanding of the logic of the functioning of the health services.

In terms of the analysis of the model with respect to the various months, it was seen that weekdays in the months of July, August and September corresponded with negative coefficients. Whereas, the months of January and February had the highest positive coefficients. This situation is due to the fact that these are the coldest months in Jaén and usually coincide with the emergence of common diseases such as colds and flu.

Air quality was observed to have a positive weight of 0.03 for the days with bad air quality and −0.03 for days with good air quality.

Finally, weather attributes also had an important weight in the model, reaching the highest weights of maximum, minimum and mean temperatures, and relative humidity.

As expected, it was confirmed that the selected algorithms performed very well when there was high temporality in the sample data. When the sample data size was small SVM performed better, as is the case for the prediction for weekends and holidays. In order to confirm these results, they were contrasted with two other regression algorithms widely used in health prediction, Random Forest and eXtreme Gradient Boosting (XGBoost) [32–35]. The theoretical accuracy of both models determined the difference between the predicted

number of patients attending the health centre and the actual data. For the weekday cohort, the results were: In the case of the Random Forest algorithm an absolute error of 6.23% was obtained, while with the XGBoost algorithm 6% was obtained.

It was confirmed that the three selected algorithms offered a higher predictive accuracy. As seen above, the GLM gave the best result with an absolute error of 2.11% compared to the 2.31% error of the SVM with a linear kernel or the 4.22% error of the SVM with a Gaussian kernel. A visual analysis of the results of the Random Forest and XGBoost algorithms is shown in Figure 11, where it can be seen that GLM fits the predictions in all months of the year.

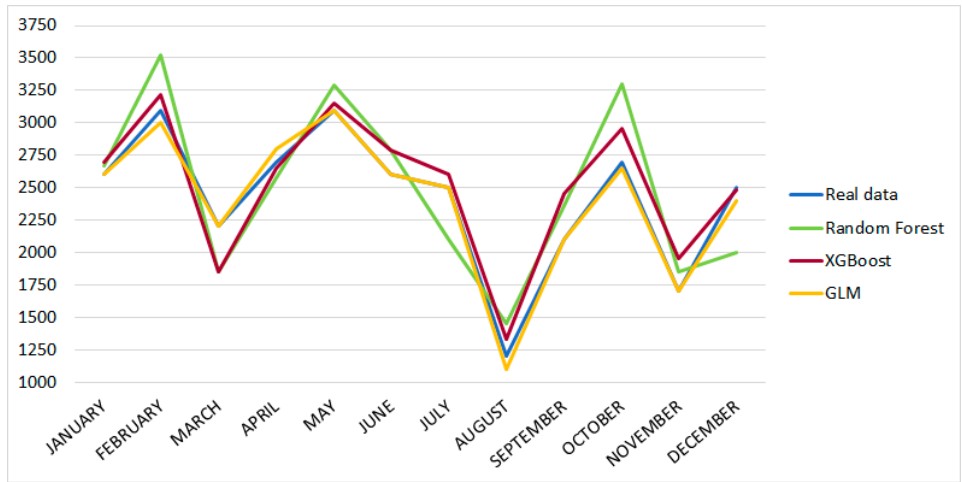

**Figure 11.** Comparison of the predictions de los modelos GLM, xgbost y random forest of the weekdays with respect to the actual data of 2018.

At the application level, the decision support system has a simple, user-friendly interface so it can be used by primary care resource managers. The decision support system can be accessed from any web browser and can be used from a PC or a mobile device. The system basically consists of a form to insert data and another final screen where the results are displayed. The main forms are shown in the Figure 12.

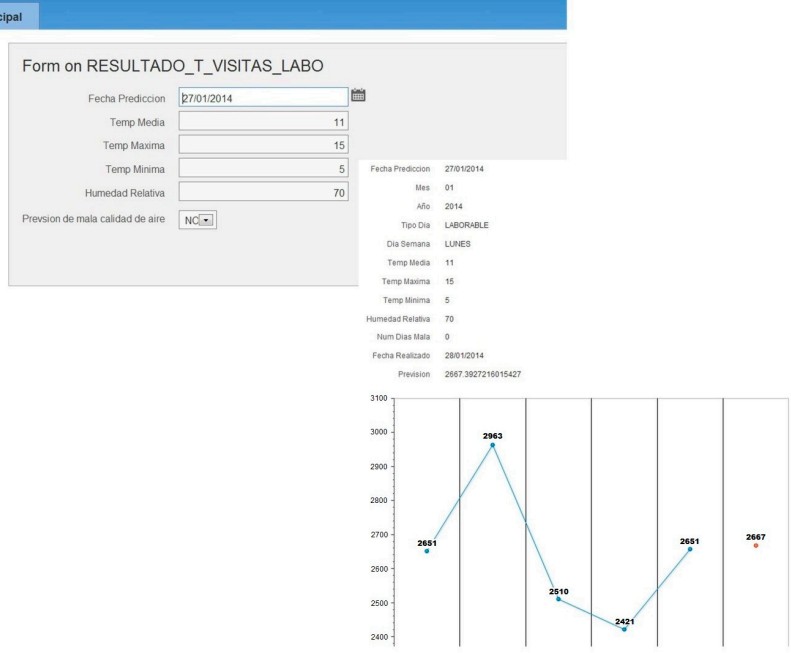

**Figure 12.** Design of the main screens of the decision support system.

Nowadays, there are many tools on the market that integrate database and data mining modules that facilitate the development of such systems. It is essential to base this work on these types of fast development tools so that research in data mining materializes in expert systems that can be used by the intended professionals. As seen in this paper, with a little knowledge of SQL and the right tools, an expert system can be easily developed.

## 5. Conclusions

The first point to highlight in this work is that a predictive model of patient flow to the primary health care centre has been designed, using a basic set of input attributes: meteorological and air quality factors. These data are available from several public web services with an early forecast of up to 10 days. More complex factors would have been an unnecessary increase in complexity for the administrator of these data values. It was proven that the results obtained by the models generated in this research with basic information were satisfactory.

The main conclusion of this work is that it is possible to develop models that are able to predict the influx of patients who need medical care with an absolute error of the 7.6% for weekend and holidays periods and 2.11% for the rest of the year, this is weekdays. In this case it was confirmed that the number of patients coming to this primary health centre followed a temporal pattern, but within this temporal pattern there was a large variation from year to year. The consequence of this was that the medical resources of this health centre were overstretched, or that the availability of the resources required were below the demand. In this sense, knowing in advance the number of patient visits to the medical centre is a key factor for decision-makers to correctly manage the necessary resources and thus optimize the medical care for citizens. In addition, from an economic point of view, this information is an indicator to health centres resources managers to estimate correctly the use of its services.

To extend this work, data from all the health centres in the city would be needed to generate a forecast of the number of patients attending each centre individually. In this way, the benefit is global, since inefficiency in the management of resources in one centre has a negative impact on another well-managed centre. Patients who come to a health centre with long waiting times go to another centre that does not have long waiting times, thus causing an imbalance in the care provided by this second centre. Therefore, the use of this tool may help to improve several aspects of health management in a city. Firstly, an optimal economic plan can be activated, if the demand for patients on a given day is known in advance, the resources of the centre can be reorganised and the centre manager can act to avoid overstaffing or understaffing. Another important aspect is that the application of the model will increase patient satisfaction, since if demand exceeds capacity and it is detected in advance, technical and human resources can be increased, thus improving the quality of healthcare service.

On the other hand, this model would also have a positive impact on emergency services, since it would reduce visits to these services for non-urgent matters, avoiding the overflow of primary healthcare services the patients tired of the long waits, and thus going to the emergency room. Finally, this model can be a very useful tool for managers of resources by providing the necessary information to be able to properly size the most efficient and sustainable primary healthcare services throughout the city.

The development of a decision support system has been achieved in a few hours without a data mining expert, as the tools available nowadays allow us to simplify these systems. The system has been developed by an Oracle developer expert in 16 h, i.e., in less than three working days. This system is now available to those responsible for managing the resources of health centres.

The decision support system is able to predict the number of patients who will seek health care in primary care centres in Jaén. This is an important tool for health resource managers, as it is available for managers to carry out simulations by introducing simple

real or simulated variables, thus contemplating different scenarios that may arise and thus developing different work teams according to the prediction data.

**Supplementary Materials:** The following supporting information can be downloaded at https://www.mdpi.com/article/10.3390/app122211453/s1.

**Author Contributions:** J.J.C.—Data processing and development of predictive models; M.I.R.—Decision support system development and article writing; F.R.F.—Reviewed the article. All authors have read and agreed to the published version of the manuscript.

**Funding:** This research received no external funding.

**Informed Consent Statement:** Not applicable.

**Data Availability Statement:** Not applicable.

**Acknowledgments:** This research has been partially funded through the research PREDIC_I-GOPO-JA-20-0006 which is co-financed with European agricultural fund for rural development and the Junta de Andalucía funds, and TIC144-Informática Gráfica y Geomática Research Group (Junta de Andalucía).

**Conflicts of Interest:** The authors declare no conflict of interest.

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
