# Peer review of "Use of Data Mining to Predict the Influx of Patients to Primary Healthcare Centres and Construction of an Expert System"

_applsci, doi:10.3390/app122211453_

Round 1

Reviewer 1 Report

The writing of this paper is not indeed good, so readers are quite hard to perceive what the authors would like to convey.  Thus, a professional proofreading service is required to improve the quality of this paper.

All decimal data in this paper is incorrectly written, which should be point instead of comma.

In fact, the main motivation for this work stated in page 2, lines [40-48], seems not to be convincing as the authors could have simpler solutions to solve their problem instead of approaching the prediction. Importantly, it happens sometimes only.

Next, if this study focuses on predicting “the number of appointments” (page 2, line 73) which can be arranged in advance, not covering the walk-in scenarios that might occur frequently in real life.

Additionally, the authors selected the data-mining techniques for their solution without providing reasoned discussions. A deep analysis could be more effective.

Similarly, there is no explanation for the idea of selecting the groups of data including season, meteorological and pollution data.

Title of section 2.2 should be “Data mining…”.

Figure 3 still displays the grammar checking.  

Why do we need to repeatedly introduce the input data in section 3.2, which has been already mentioned in previous section?

Table 1, Figures [4-5] are shown in the paper, but they are never used or explained.

Page 11, line 311, it should be “On the right side of Figure 6…”.

What are the section of “Development of the application with APEX”, and “System learning through data feedback”?

Title of section 4 is inconsistent.

Page 13, line 360, it should be “… the two cohorts, table 9”?

Page 14, line 3373, it should be “… are shown in Table 10”?

Page 16, line 423, it should be “…with an absolute error of 12.77, table 12”?

Most of the contents in the conclusion section should be placed in section 4 since the conclusion just provides a summary of the achieved results.  

Last but not least, the algorithms the authors used to do some analysis on the achieved data are outdated. Each has its own performance limitations on specific categories of the datasets. In addition, the relationship between the environment and the influx of patients found is insignificant for the purposes of understanding the logical functioning of the health services as analyzed and described in the paper.

Author Response

Tha Author´s reply to the review report is in the file uploaded.

Reviewer 2 Report

This manuscript introduces of using data mining to analyse the data and predict the influx of patients who will attend the primary health care centres in District of Jaén. This study is important for addressing the design of data mining algorithms and development of an expert system. The core idea seems interesting, but the paper should be improved in some regards:

What is the main motivation of this study? What will the gap be addressed here? There are various data mining models that have been studied and used by various researchers. What is the framework proposed here of using data mining for prediction? More explanation about the proposed technique/modal should be given. Please explain a few more works that are related to improving the health system using data mining tools for prediction. What is the significant difference between this manuscript with other works?

Briefly describe the model validation (2018). How to perform the validation. Clearly state which performance metrics are used and its formula. Figure 3 should be explained clearly on how to develop the expert system. Please briefly explain the dataset & the scope of the study by choosing primary health care centres in the District of Jaén.

Abstract should be improved. Should describe in more detail the proposed technique. What type of data mining algorithms are used here? Also briefly explain the design and development of the system. What are the findings and conclusions of this manuscript? Please revise the abstract.

Please avoid usage of first-person pronouns such as “I,” “We,” “ours,” “us” as much as possible to maintain the tertiary nature of this publication and maintain a neutral voice in the article. For instance, 'We' can be rephrased as 'the authors'.

Sections 4 & 5 should be improved. Some are redundance and repeated the same explanation. It would be better to see more implication and justification of the findings for all the analysis & results rather than just reporting the results. Should highlight the conclusion/significant contribution and impact of this study.

Lines 24-29: Briefly describe these unclear/confusing sentences.

Line 57: What is COPD? Spell it out for the first-time acronym. Need to be consistent with the use of acronyms, elaborating the acronym first before using the acronym. Please check for other abbreviations, e.g., REDIAM, RDBMS, SGBD, CDM, GLM, SVM, etc.

Line 76: Who are the ‘authors’ here?

Line 78: “…high level of precision”. Is it?

Line 96: “…using data mining techniques [23-27]…”. Please explain which data mining techniques are used in this manuscript. Please justify the selection. Any latest references regarding data mining? How is your research different from other prior works published? More latest references are required throughout the manuscript.

Figures 1 & 2: Would be better to clearly state that 2014 instead year 1 & so forth.

Table 1: Should mention and describe in the text.

Line 160: should be 12 is December. (same for line 224)

Line 165: should be Data.

Line 257: Should be Table 4 instead of Table 5. Prefer to use ‘dot’ instead of ‘,’ for the weight number (e.g., 0.34). More description should be given to this table. Please check the sequence number of other tables and figures.

Figure 4 should be explained in detail. Almost zero for air quality? In which year? Same for Figure 5. For captions, it should also be for weekends. Line 303: why is GLM selected for working days and SVM for non-working days? Please explain & justify.

Line 388: Figure 8 should be explained in detail. Is it wrong caption for SVM? Should be GLM.

Table 11 should be explained in detail. Should be mentioned in the text. Should we arrange the attribute (e.g., month) in ascending for ease of comparison?

Line 423: Should be Table 12. Captions should include weekends. It should be explained in detail. Why for this cohort, GLM is not the best algorithm? Please justify.

Line 438: Is this Figure 9 like Figure 8?

There needs to be a thorough copyediting done of the manuscript will require proofreading and editing. There are some repetitions in the paper, please thoroughly revise the paper for improving the readability. Grammar/English/Typo Errors are:
Line 126: 20217 should be 2017.
Lines 128-129: Confuse sentence. Summer and winter? Which months?
Line 169: 2017 y 2018?
Line 175: figure 3 should be Fig. 3. Should describe more about Fig. 3.
Line 272: Please check this sentence.  
Line 280: lineal should be linear
Line 311: Figure 4 should be Figure 6.
Line 316: Should be Table 5
Line 319: Check this subsection numbering.
Line 337: Should be Table 6
Line 344: Should be Table 7
Line 348: Check this subsection numbering.
Line 353: Should be Table 8
Line 357: Check the font size of Section 4.
Line 360: Should be Table 9. Please check if this Table is the same as Table 4?
Line 373: Should be Table 10 (or 9). Caption Table 10 should be Table 9? Please check for consistency. Please also explain the performance metrics used in this table. Why select these three models, SVM with Gaussian Kernel, SVM with Linear Kernel, & GLM?
Line 383: 2.31 should be 2.31%
Line 439: Should be Table 13
Line 470: Should be Figure 10
Line 480: Should be 5. CONCLUSIONS

The revisions would help enhance the quality of the work.

Author Response

The Author's Reply to the Review Report (Reviewer 2) is in the file uploaded

Reviewer 3 Report

Detailed points are included as comments in the appended PDF. The following is a summary:

General:
----------
(i) I would the call system a "decision support system" rather than an "expert system".
(ii) After building the predictive models for patient influx, but then the conclusion is very generic (can help..., can be a useful tool). I would include a couple of specific real "use cases" where the prediction would be as decision support for logistics/resource assignment, with reference to a real hospital. So, an "end user story" of a medical coordinator, how s/he would consult the system, and how they would then use the information to perform actions.
(iii) Overall the inclusion of figures is quite messy (centered and aligned with the text) which makes it difficult to follow, this needs to be improved. Also, some section structure/subheadings are not clear. For example, page 11, "1- Development of the application with APEX" in bold comes after "3.3-. Generating the Models with Oracle Data Mining". I would eliminate the former).

Overall: The quality of the presentation (graphics, include some material in an appendix) needs to be significantly improved. Also, there appear to be some mistakes in the conclusions (about which algorithm gave best results) which don't correspond to the empirical results. In the testing of the predictive models, it is not clear what data is used to train the models, and which to test. The utility of the predictive models for hospital managers treated very generically in the conclusions. I would expect a couple of real "use cases" where the end user asks the system for a prediction, and then uses that prediction in an actionable way (e.g. x medics required with specialty y, z beds required from date A to date B, etc.).

Abstract
----------
"the study is based on historical attendance parameters, meteorological data and air quality"
the meteo and air quality data are specific to the study focus to relate patient attendence to these factors. However, in the data plots and analysis, the term "holidays" is used for the summer which is not very scientific in my opinion, if you are considering meteo and air quality data due to climatic/seasonal variations. If you consider "holidays/vacations" as a factor, it could just mean less people (in Jaen) visit the clinic/hospital because they (i) have left Jean and gone to another place for vacations or (ii) people prefer to enjoy vacations rather than visit the clinic/hospital. Also would need to distinguish between routine/non-urgent clinic visits and urgent ones, which might have a different tendency.

Intro
------
"However, this research has remained at the theoretical level. Complex, user unfriendly programs must be used to make a prediction, and only their developers know how to operate them. As a result, these systems are not used."
This seems a very big generalization of the state of the art. I would expect there are some user friendly applied systems in this area.

"it is essential to have an expert system with a user-friendly interface"
Rather than an "expert system", I would call it a "decision support system".

"However, these developments are often not implemented due to their high cost and complex development."

I don't agree much with this statement, a user interface can be developed in Javascript for a web app, and adequate end user screens implemented, for a reasonable price, with an averagely skilled professional programmer. However, this also depends on the "back-end" providing the right data/information to the "front-end".

Figures/plots in general: should be improved to look less like standard Excel plots. Also, some figures have tabular data included (e.g. figs. 1 and 2) which should be put in a separate table. Furthermore, some plots can be "scatter plots" with a trend curve estimated, rather than just stratight lines joining the points.

Pages 7 through to 13 there are several code  segments which should go in an annex. The same figs. 6 and 7 which could go in annex (fig. 7 is not very relevant for an academic paper).

Page 19, fig. 10: You mentioned at the beginning that having a user friendly user interface was important. But this seems to be a standard data entry form, I would say some more effort could be done to make it more user friendly, such as visual assets (icons, plots of key variables which shows max/min/avg graphically, and include the standard deviation).

Conclusion
-------------
I think it was the SVM with the Gaussian Kernel which gave the best result (minimum error) in both cases (Tables 10 and 12). So I think the best techniques you mention in the conclusion do not correspond to the best techniques found in the empirical testing.

Author Response

The Author's Reply to the Review Report (Reviewer 3) are in the file uploaded

Reviewer 4 Report

This paper aims to utilize existing machine learning algorithms to predict the number of patients that need medical attention. The paper is well-written. However, the novelty level needs to be pushed significantly to meet the journal level. Neither a new computational method nor extensive experimental study against baselines is presented. The paper just employs  machine learning algorithms (e.g., GLM and SVM including two kernels), which is typically performed in a data mining class.

Comments (to improve the paper):
-Making a web server for tool availability purpose
-Including a performance comparison against baselines using benchmark datasets
-Employing XGBOOST and Random Forests to report additional performance behavior
-Utilizing a statistical test to report the method that yields significant results
-Designing new learning algorithms to improve the prediction performance is a plus

Author Response

The Author's Reply to the Review Report (Reviewer 4) is in the file uploaded

Round 2

Reviewer 1 Report

Thank you for the authors' effort. 

Please consider my comments below: 

The revised version of the paper should be submitted along with the tracking file.

The authors’ responses should be numbered, and they still have many writing problems (e.g., grammar).

Figure 3 still displays the grammar checking INSIDE the figure. 

The paper should also describe the future work.

The contributions of this paper are indeed trivial in general, and the algorithms proposed are rather naïve, which should be used as baselines. 

Reviewer 2 Report

The authors had put a lot of effort into improving the manuscript. However, some of the comments raised previously have not been addressed properly and correctly. For example,

More explanation about the proposed technique/modal should be given. Please explain a few more works that are related to improving the health system using data mining tools for prediction. What is the significant difference between this manuscript with other works?

How to perform the validation. Clearly state which performance metrics are used and its formula. Figure 3 should be explained clearly on how to develop the expert system.

The manuscript lacks originality/novelty and significance contributions. 

Reviewer 4 Report

Comments raised in a previous round of review have not been addressed. requested learning algorithms (i.e., XGBOOST and Random Forests) have not been utilized, p-values were not reported and other points have not been addressed.

Round 3

Reviewer 2 Report

The authors had considered and addressed most of the comments. The manuscript has improved and could be considered for publication after final proofreading and improvement.

Author Response

The authors thank the reviewer for his interest and dedication to improve the quality of this manuscript.

Reviewer 4 Report

1 comment "-Employing XGBOOST and Random Forests to report additional performance behavior" was addressed out of the 5 comments in the first round of review. Highly recommending to increase the novelty level. Just evaluating the performance using existing machine learning algorithms applied to a dataset (also not a benchmark one) doesn't meet the journal level

That is my objection
